# Novel Approach to Pharmaceutical 3D-Printing Omitting the Need for Filament—Investigation of Materials, Process, and Product Characteristics

**DOI:** 10.3390/pharmaceutics14112488

**Published:** 2022-11-17

**Authors:** Thomas Pflieger, Rakesh Venkatesh, Markus Dachtler, Karin Eggenreich, Stefan Laufer, Dominique Lunter

**Affiliations:** 1Digital Health Systems GmbH (DiHeSys), 73529 Schwaebisch Gmuend, Germany; 2Gen-Plus GmbH & Co., KG, 81477 Munich, Germany; 3Pharmaceutical Chemistry, Eberhard Karls University, 72074 Tuebingen, Germany; 4Pharmaceutical Technology, Eberhard Karls University, 72074 Tuebingen, Germany

**Keywords:** pharmaceutical three-dimensional printing (3DP), hot-melt extrusion (HME), printability, oscillatory rheology, novel printhead design

## Abstract

The utilized 3D printhead employs an innovative hot-melt extrusion (HME) design approach being fed by drug-loaded polymer granules and making filament strands obsolete. Oscillatory rheology is a key tool for understanding the behavior of a polymer melt in extrusion processes. In this study, small amplitude shear oscillatory (SAOS) rheology was applied to investigate formulations of model antihypertensive drug Metoprolol Succinate (MSN) in two carrier polymers for pharmaceutical three-dimensional printing (3DP). For a standardized printing process, the feeding polymers viscosity results were correlated to their printability and a better understanding of the 3DP extrudability of a pharmaceutical formulation was developed. It was found that the printing temperature is of fundamental importance, although it is limited by process parameters and the decomposition of the active pharmaceutical ingredients (API). Material characterization including differential scanning calorimetry (DSC) and thermogravimetric analyses (TGA) of the formulations were performed to evaluate component miscibility and ensure thermal durability. To assure the development of a printing process eligible for approval, all print runs were investigated for uniformity of mass and uniformity of dosage in accordance with the European Pharmacopoeia (Ph. Eur.).

## 1. Introduction

Recent advances in additive manufacturing including 3D printing have undoubtedly had a major global influence on technology across different fields [1]. Numerous drug delivery systems and devices in the medical and pharmaceutical sector are already being successfully printed in a research environment. In regard to the pharmaceutical industry, additive manufacturing offers the option of extensive medication customization [1,2,3,4]. Integrating additive processes has several advantages over the current, well-established but outdated and rigid “one size fits all” approach that provides limited flexibility in dosage tailoring [1,3]. Aside from the two major advantages of avoiding medication errors and enabling a flexible treatment to the patient, printing tailored oral dosage forms is financially attractive for both costly medications as well as small scale on-demand production [3,4,5]. The development of dosage forms, production of sample batches, and modification of samples can be done with little effort compared to generic powder-pressed tablets or filled capsules which require heavy pharmaceutical machinery [4,5]. Alongside the mentioned advantages of pharmaceutical 3DP, the technology enables prompt assessment of feasibility and applicability of highly complex dosage forms and devices. Recent research efforts reveal that innovations such as personalized fluoride-eluting mouthguards, individualized nasal piston devices, compartment-, combi-, core-, shell-, alternating-, bi-, or poly-pills are solely possible thanks to advances in pharmaceutical 3DP technology [6,7,8,9,10].

Currently, the majority of drug formulations have a predetermined dosage of one or more active pharmaceutical ingredients (API) [4,11]. This does not adapt to the physiological constitution of the patient. Dosing drugs accurately depends on genetic, metabolic, and gender-specific properties as well as disease state [3,12]. Highly potent drugs in particular have a narrow therapeutic window, which varies from patient to patient. This calls for an individual treatment through personalized healthcare. Non-tailored drugs fail to meet the requirements for treating patients immaculately and there is the possibility of dosing inaccuracy [13].

Hypertension (HTN) is a medical condition in which blood pressure is persistently elevated. Long-term high blood pressure levels lead to an increased risk for cardiovascular and renal complications [14]. Therapy for stage one hypertension incorporates β1-receptor blockers such as the model drug Metoprolol Succinate (MSN) [14,15]. In practice, the administration of hypertension reducers is implemented by a stepwise controlled care approach to reach target levels of blood pressure improvement [16]. During the course of the treatment, various medications with different doses are administered [14,16]. This is where pharmaceutical on-demand printing comes in handy. 3DP technology offers the possibility to customize dosage forms in coordination with the therapeutic progression [1].

The 3D printhead used in this work utilizes a special functional principle. The Flexdose^TM^ printer (FDP) developed by DiHeSys: Digital Health Systems GmbH (DiHeSys) is an extrusion-based printer, whereby the feeding is achieved by granules. The granules comprise pharmaceutical polymer formulations that hold particular active ingredients and are prepared by table-top HME starting from powder formulations. This FDM printhead gradually builds up three-dimensional oral dosage forms by dispensing polymer melt through an extruder nozzle in horizontal layers. FDM emerged to the most used 3D printing technique in pharmaceutical research and development due to its economic acquisition cost and reasonably low equipment and setup requirements [17,18,19]. This technology offers extensive design freedom, the ability to realize complex structures, and rapid prototyping [17,18]. The printing success is based on a sophisticated interplay between model design, hardware, process parameters, and polymer formulation [17,19].

To overcome the filament strand limitation for pharmaceutical fused deposition modeling (FDM), alternative extrusion methods such as powder and pellet direct extrusion have come into focus just recently [20,21,22]. As the literature for filament-fed printheads exhibits, the fabrication of suitable filament materials is difficult [18,23,24]. The strand must fulfill precise mechanical requirements to be able to be fed regularly by gear wheels and the printer only allows small deviations in the filament diameter [18,25]. These problems do not arise with the granules system. To ensure continuous extrusion, the complex melt viscosity of the viscoelastic polymer granules must be examined and adapted to the specific properties of the extrusion channel design [26].

The aim of this study was to investigate crucial process parameters for DiHeSys’ novel granules-fed pharmaceutical Flexdose^TM^ printer emphasizing melt rheology of printable MSN polymer systems and characterization of product printlets. Regarding the final drug delivery systems, an immediate release (IR) of the model drug is aimed for. Therefore, the pharma-grade IR polymers KVA64 and EPO were utilized. These include a co-povidone polymer with an erodible instant release matrix and a methacrylate co-polymer that is soluble in acidic media, respectively. The polymers’ solubility attributes suit the in vitro dissolution method applied in this work.

## 2. Materials and Methods

### 2.1. Materials

The model substance used was the hypertension drug Metoprolol Succinate (MSN) with purity >98% supplied by Hangzhou Longshine Bio-Tech Co., Ltd., Hangzhou, China. The polymers Kollidon^®^ VA64 (KVA64) and Eudragit^®^ E PO (EPO) were kindly donated by BASF Pharma SE, Ludwigshafen, Germany and Evonik AG, Essen, Germany, respectively. The pharma grade plasticizer Lipoxol^®^ 6000 (PEG) was obtained from Sasol Chemicals AG, Johannesburg, South Africa. All chemicals used were of analytical grade and used as received.

### 2.2. Methods

#### 2.2.1. Formulation of Blends

The formulations consist of a single carrier and diluent polymer KVA64 or EPO, model drug MSN and plasticizer PEG optionally. The formulation compositions can be found in Table 1. The blends represent the final formulations that were used for additive manufacturing. The batch size for each of the four formulations was 50 g. Blend components were pre-weighed and three-step geometrically mixed at 49 rpm for 15 min in a Turbula^®^ T2F tumble mixer from WAB Group AG, Muttenz, Switzerland.

#### 2.2.2. Production of Granules

The prepared physical mixtures underwent twin-screw hot melt extrusion with lab scale extruder ZE HM9 from Three Tec GmbH, Seon, Switzerland. The module comprises co-rotating elements with a die diameter of 2 mm. Table 2 shows extrusion temperatures, torques, and screw speeds set and obtained for HME of each blend. The extrusion channel consists of three equivalent temperature zones. The extrusion screws are solely conveying screws and have no kneading elements. Since the filament diameter of the output is irrelevant for the production of granules, it has not been monitored. The extrudates were kept in sealed plastic bags to avoid moisture sorption. The extrudate strands were downsized to granules through rasp sieve milling with a U5 Comil^®^ from Quadro Engineering Corp., Waterloo, Canada, at 250 rpm. Granules with maximum diameter of 2 mm were obtained. Granules less than 0.6 mm in size were separated using a stack sieve.

#### 2.2.3. Printlet Design and 3D Printing Process

Tablets were fabricated by 3DP using the pharmaceutical 3D extruder printer from DiHeSys GmbH, Schwaebisch Gmuend, Germany. Figure 1 shows the biplane tablet printed in this work. The stereolithography template (r = 6.00 mm; h = 6.00 mm; V = 0.679 cm^3^) was sliced into g-code (.gcode) with Ultimaker^®^ Cura 4.10.0 by Ultimaker B.V., Utrecht, Netherlands. For printlet mass scalability and mass uniformity studies, the mentioned stereolithography template was additionally scaled to a volume of 75% (r = 5.45 mm; h = 5.45 mm; V = 0.509 cm^3^) and 50% (r = 4.76 mm; h = 4.76 mm; V = 0.339 cm^3^). Tablets were printed using standardized settings as follows: fine resolution slicing; extrusion factor 1.131 mm^3^/s; speed factor 25 mm/s; wall thickness by three circumnavigations; 100% body infill; no base brim, supports, or rafts; build plate temperature of 50 °C. The nozzle temperature was set according to the respective formulation as stated in Table 2.

#### 2.2.4. Printability Runs

A print run is considered successful under the requirement of continuous polymer melt extrusion without major print defects, clogging, polymer melt backflow, or generation of printer casing damaging backpressure for at least 24 min (3 × 8 min). This is the time required to print three standard 100% body infilled biplane tablets at standard extrusion and moving speed. The printlet shows no free spaces, air hollows, irregularities, warping, sharp edges, under- or over-extrusion, offsets, stringing, or other undesirable effects.

#### 2.2.5. Differential Scanning Calorimetry (DSC)

The DSC studies were carried out on a DSC 1 from Mettler Toledo, Columbus, OH, USA, using 100 μL aluminum crucibles with 8–15 mg of sample in duplicates. Working conditions covered a range of 25 °C to a maximum of 175 °C with a heating rate of 10 °C/min, under nitrogen atmosphere with a flow rate of 30 mL/min. The tests were performed on physical mixtures and granules to follow the API physical state conversions along different processing steps.

#### 2.2.6. Thermogravimetric Analysis (TGA)

The proportional weight loss was determined by a STA 409 PC/PG Luxx^®^ from Netzsch GmbH, Selb, Germany, in nitrogen atmosphere (flowrate 20 mL/min) from 30 °C to 230 °C with a heating rate of 10 °C/min. Duplicate tests were performed on unprocessed substances, blended physical mixtures, and granules in order to choose processing temperatures that would not result in harmful degradation effects. All samples were measured no later than one day after production.

#### 2.2.7. Rheology: Small Amplitude Oscillatory Shear (SAOS)

The SAOS tests were performed with a Physica MCR301 Rheometer from Anton Paar GmbH, Graz, Austria, in oscillation mode with parallel plate configuration. Rheological measurements exclusively involved extruded blends. Samples were placed on a pre-heated Peltier plate, melted, and compressed to a 1.0 mm gap by a 25 mm diameter disposable stainless steel plate. The measurements were performed within the linear viscoelastic region (LVR), established by strain sweeps executed at the minimal processing temperature. Strain sweeps were conducted from 0.01% to 10.0% strain at 10 rad/s angular frequency. Consequently, frequency sweeps were performed within the LVR range at decreasing angular frequencies from 500 to 1 rad/s as to determine material viscoelastic behavior in relation to time and frequency. The rheological evaluations were carried out in duplicates.

#### 2.2.8. Uniformity of Mass of Single-Dose Dosage Forms

For the development of an applicable process, a high degree of printlet mass uniformity is crucial. In accordance with the European Pharmacopoeia 2.9.5 “Uniformity of mass of single-dose dosage forms”, ten single oral dosage forms (ODFs) were printed with formulations KVA64/PEG/MSN and EPO/MSN [27]. For ODFs with a mass of more than 250 mg, it is required that not more than one of the individual masses deviate from the average by more than 5% and none deviates by more than 10%. For each preparation, 20 tablets were weighed individually and the arithmetic mean masses and standard deviations were calculated following the mentioned monograph. 

#### 2.2.9. Evaluation of Content Uniformity

The uniformity of dosage units was evaluated according to the European Pharmacopoeia 2.9.40 “Uniformity of dosage units” [28]. Regarding the monograph, the acceptance value (AV) was calculated for each batch. The AV is required to not exceed a value of 15 (limit 1; L1). If the requirements for the first testing level (*n* = 10) are not met, 20 additional dosage forms need to be evaluated and the total AV is not allowed to exceed 15. In addition, no single dose may deviate from the reference value by more than 25% (limit 2; L2). For the sampling of 3DP tablets, 10 separate specimens were used and measured using the analytical method described below.

The preparations were crushed using a mortar and pestle. The samples were mixed with acetonitrile, agitated for 12 h and filtered through a 0.45 μm filter from Millipore Ltd., Dublin, Ireland. The individual contents of model drug were determined by UV-HPLC Agilent 1200 series from Agilent Technologies Inc., Santa Clara, CA, USA. The eluent was screened at a wavelength of 223 nm. The method showed linearity between 1 and 280 mg/L with R^2^ = 0.99996 under the same conditions. The limit of detection (LOD) and the limit of quantitation (LOQ) for MSN were estimated to be 0.06 mg/L and 0.10 mg/L, respectively. 

#### 2.2.10. In Vitro Dissolution

Determination of the in vitro drug release was performed using USP type II dissolution apparatus Sotax AT7 from Sotax AG, Basel, Switzerland, in 900 mL of 0.1 M hydrochloric acid at 37 °C with a paddle speed of 100 rpm. Sampling was executed every 5 min for the first 20 min, every 15 min of the following 2 h, and continuing with every hour up to 4 h. Dissolution studies were performed in triplicate and the average proportional cumulative drug release was plotted as a function of time. The MSN concentration in the dissolution medium was measured using a HP 8453 UV-Vis Spectrophotometer from Agilent Technologies Inc., Santa Clara, CA, USA, at a wavelength of 223 nm in a 1 cm cell versus a blank solution consisting of 0.1 M hydrochloric acid. The applied calibration range was between 1 and 280 mg/L. The LOD and LOQ were found to be 0.17 mg/mL and 0.50 mg/L, respectively (R^2^ = 0.99997).

## 3. Results

### 3.1. Thermogravimetric Analysis

During the production process of 3D printed dosage forms, the substances are exposed to elevated temperatures in two separate processes. These include tabletop extrusion to produce drug loaded granules and the actual 3D printing process. Shear forces, inner friction, and other temperature increasing deviations can occur during these process steps. Thus, all formulations are supposed to be processed below the decomposition temperatures of the pure API and at lowest possible processing temperatures in general. Since temperature degradation of the API and other excipients must be avoided during all process steps, TGA measurements were performed. Samples are considered to be thermally stable up to an accumulated gravimetric mass loss of 3%.

The degradation temperature of pure drug MSN was found to be 177 °C (Figure 2a). The sample did not show water evaporation, which leads to the assumption that the drug batch is dry. The placebo systems KVA64/PEG and EPO are thermally stable over the entire observed temperature range up to 230 °C (Figure 2b). The initial mass plateau drops are attributed to the loss of adsorbed water in both cases. While EPO lost 0.6 wt.%, KVA64/PEG lost 2.4 wt.% adsorbed water. This trend also continues with physical mixtures and extruded granules. The KVA64-based systems draw more moisture than the EPO-based blends. Starting from the plateau, after complete moisture evaporation, KVA64/PEG/MSN and EPO/MSN blends show decomposition temperatures of 212 °C and 206 °C pre-extrusion, respectively (Figure 2c). While EPO/MSN granules show a comparatively low water content of 0.72 wt.%, KVA64/PEG/MSN granules’ water content increased to 2.9 wt.%. The results indicate that the additional HME processing step influences the samples by making them more vulnerable to water absorption. Since the samples were transferred directly into sealed plastic bags after extrusion, the moisture sorption must have occurred during HME processing. Starting from the dry plateaus, KVA64/PEG/MSN and EPO/MSN granules are thermally stable up to 207 °C and accordingly 206 °C (Figure 2d).

### 3.2. Differential Scanning Calorimetry

DSC data of formulations KVA64/PEG/MSN, EPO/MSN and according single blend components are shown in Figure 3 and Figure 4. Raw model drug MSN’s melting point was found to onset at 138 °C, as expected (Figure 3a) [29]. All formulations that contain MSN indeed display this melt peak to a certain extent (Figure 3d–f and Figure 4c–e). This indicates that MSN is partially present in crystalline form in both polymer matrixes no matter if post- or pre-HME [30]. As no amorphous solid dispersion was aimed for, this result is acceptable. It should be mentioned that exothermic events only occur during the analysis of EPO/MSN granules and 3D extrudates (Figure 4d,e). The KVA64-based formulation shows no exothermic events (Figure 3d–f).

#### 3.2.1. KVA64-Based Formulation

In KVA64PEG/MSN samples, model drug melting point depression occurs increasing with processing step progress (Figure 3d–f). From physical mixture to post-print tablet, the intensity of the MSN and PEG melting signals decrease. This indicates progressively enhanced solution of these two substances in the carrier polymer. Nevertheless, the model drugs saturation limit seems to be reached. Due to desorption of water, raw polymer KVA64 has a broad endothermic peak in the moderate temperature range below 100 °C (Figure 3c), which cannot be found to the same extent in processed samples.

#### 3.2.2. EPO-Based Formulation

MSN has reached its solubility capacity in EPO, since the intensity of the model drug melting peak remains identical across all processing steps (Figure 4c–e). EPO/MSN granules’ DSC data at 90–125 °C is particularly remarkable (Figure 4d). While the actual MSN melting peak persists, endothermic and exothermic events occur in this area in a narrow temperature range in direct succession. After processing the granules into tablets, this area changes into two clear single thermal events almost identical to the thermal events in the physical mixture (Figure 4e). By reason of low water content in pure EPO, no water desorption can be detected, only a weak endothermic signal of unknown origin (Figure 4b).

### 3.3. Small Amplitude Oscillatory Shear Rheology

#### 3.3.1. Technical Challenges Regarding Melt Viscosity

There are two main viscosity-related issues causing damage to the extrusion channel or ceasing material output which can be understood with rheology measurements. For continuous extrusion, the already mentioned qualitative effects “clogging” and “polymer melt backflow” must be avoided by controlling the polymer melt rheology. The term clogging describes highly viscous polymer melt building up a clog alongside the extrusion channel that blocks polymer conveying [26,31]. It leads to elevated torques, backpressure, and risk of jamming [26,31]. Polymer melt backflow is an undesirable effect in HME where low-viscous polymer melt is migrating in direction of the channel top instead of the nozzle [32]. Chiruvelle et al. showed that the aforementioned flow phenomenon is strongly linked to the viscosity of the polymer via the energy equation [33]. The viscosity of non-Newtonian fluids is a function of temperature and shear rate [32,33]. Polymer melt backflow restrains polymer conveying, extrusion, and therefore material output.

#### 3.3.2. Establishing Target Rheological Properties with Placebos

To establish a first estimation of target rheological parameters, printability runs were performed with the placebo formulations at different temperatures. The aim was to identify a relation between rheological properties of a polymer system and its printability. A formulation was considered printable if it exhibited continuous extrusion of the polymer melt without substantial print errors, clogging, polymer melt backflow, or formation of casing damaging backpressure for 3DP of at least three tablets. The printability was correlated to the rheological properties of the melts.

Shear rate dependent complex viscosity results were gathered for placebo formulations KVA64/PEG and EPO. Four measurements each at different temperatures of 120 °C, 140 °C, 160 °C, and 180 °C were recorded (Figure 5). KVA64/PEG and EPO both show flat viscosity curves at each measuring temperature, which indicates high shear rate independence at all temperatures. With increasing temperature, the complex viscosity levels decrease as expected. At 180 °C, KVA64/PEG’s shear rates above 80 s^−1^ could not be monitored due to the polymer melt being highly liquid. The sample did not remain in between the rheometer’s parallel plate setup.

Printability tests showed that for KVA64/PEG, there was no extrusion possible at 120 °C nozzle temperature due to clogging. The extrusion channel jammed because of a highly viscous and hardened polymer clog. The effect occurred instantly and at no point was polymer melt obtained from the printing nozzle. The same effect occurred with EPO’s print run at 140 °C nozzle temperature.

At 180 °C nozzle temperature, no extrusion of KVA64/PEG was possible either. In this case, the polymer melt showed inapplicability regarding low viscosity. From a technical point of view, the driving force of pushing a polymer melt through the nozzle is the generation of pressure in flow direction caused by the conveying of polymer. At 180 °C, generation of backpressure occurred, and polymer melt flowed into the direction of the channel top instead of being pushed to and out of the nozzle.

With nozzle temperatures of 140 °C and 160 °C for KVA64/PEG and 180 °C for EPO, flawless printability was observed. All necessary parameters and requirements for continuous extrusion were met. Therefore, the three viscosity curves serve as reference points for the viscosity of future polymer formulations and as soft limits. A viscosity between 20 and 100 Pa·s seems to be suitable for 3DP. The study shows that the printing temperature, in strong correlation to the viscosity of a material, is a crucial parameter for 3DP application. This was verified with drug–polymer mixtures.

#### 3.3.3. Transfer to Drug Loaded Formulations

Rheology results of blend KVA64/PEG/MSN were compared with the target regime of the placebo KVA64 (Figure 6). It is noticeable that the complex viscosity window of KVA64/PEG/MSN is tight compared to that of the placebo. Small changes in temperature resulted in relatively strongly deviating viscosity results.

At 160 °C, taking a measurement with KVA64/PEG/MSN was not possible due to it reaching an almost water-like low viscosity. The measurements taken at 120 °C and 140 °C show flat gradients and therefore high shear rate independency similar to placebo KVA64/PEG (Figure 6).

The printability test runs showed that for KVA64/PEG/MSN at 120 °C, extrusion was possible in principle, but it was discontinuous and fragile (Figure 6). At increased temperature of 140 °C, a smooth print was achieved. At 160 °C, KVA64/PEG/MSN’s low-viscous polymer melt migrated backwards, no extrusion was realized, and the printhead including the extrusion channel was physically damaged because of hardened polymer at the tip of the channel.

When investigating EPO/MSN mixtures, it can be seen that MSN has a strong plasticizing effect (Figure 7). Compared to placebo EPO, lower viscosities are observed at the same temperatures and even small increases in temperature have a strong influence on the viscosity. In the case of the drug-loaded system EPO/MSN, there are comparatively large differences in viscosity, especially between the measurement temperatures of 120 °C and 140 °C.

In Figure 7, almost congruent viscosity curves were obtained for the samples EPO at 140 °C and EPO/MSN at 120 °C. Successful printability could not be achieved at either temperature. In both cases, the printhead’s extrusion channel clogged. There was no successful 3DP for placebo EPO even at 160 °C, since stringent torque prevented extrusion. At elevated shear rates, a rheological investigation of EPO/MSN at 160 °C was not feasible due to the sample leaking the rheometer’s measuring gap. The polymer melt showed severe low-viscous behavior. Smooth prints were achieved with samples of EPO at 180 °C and EPO/MSN at 140 °C fitting previously gained results for printability.

### 3.4. Uniformity of Mass

There are several fundamental approaches to deliver tailored drug doses through 3D printing [34]. Dosage flexibility is realized by altering the number of printed layers, printing multiple objects, adjustment of feed drug loading, or modification of tablet volume [34,35,36,37,38]. In this study, printlets of three different volume scale factors were examined to assess the approach’s eligibility for customized drug administration.

The results of the mass variation evaluation are presented in Table 3. For both formulations, the same print file and print settings were used. Theoretically expected printlet masses calculated for the assumption of perfect full printlet infill with proportional true densities of the blends’ single components are stated for comparison as well. Both formulations display different densities due to their differing composition. For full printlet infill, the true densities of the blends’ yield ϱ(KVA64/MSN) = 1.192 g/cm^3^ and ϱ(EPO/MSN) = 0.906 g/cm^3^. Consequently, a higher mass is to be expected for KVA64/PEG/MSN tablets than for EPO/MSN tablets when printing an identical 3D object.

Table 3 shows the results of the mass uniformity test. A certain amount of printlets exceeded the first specification limit but the secondary pharmacopeial specification limit was not surpassed in any of the print runs. Thus, all printing processes met the requirements of the Ph. Eur. monograph for single oral dosage forms [27]. The obtained data show that a process eligible for approval is feasible for both polymer systems.

Regarding standard deviations in Table 3, formulation KVA64/PEG/MSN displays a trend worth mentioning. With increasing printlet volume, the standard deviation within the print runs rises. Consequently, the printing process becomes less precise as the printlet volume increases. This trend cannot be transferred to EPO/MSN. The highest mass uniformity is obtained with a volume scaled to 75% in this case. The printing process of EPO/MSN is superior to that of KVA64/PEG/MSN in terms of printlet mass reproducibility, especially for tablet volumes scaled to 50% and 100%.

### 3.5. Printlet Volume–Mass Correlation

For an ideally scalable printing process, there is a linear correlation between the tablet mass and the volume of the tablet print file. In this case, by adjustment of the printlets volume, altered masses and thus amounts of pharmaceutical ingredient can be realized in an individualized and reproducible manner. Figure 8 illustrates the relation of printlet masses to volume scale factor. Complete scalability linearity is not achieved with either formulation (Figure 8). With blend KVA64/PEG/MSN, the obtained printlet mass offsets at 100% volume compared to the other points. With this formulation, the process comes closer to the target tablet mass especially with large printlets. Further optimization of the nozzle throughput can be addressed in multiple ways by adjusting print parameters such as the printing temperature, print speed, screw speed, or layer height [26,39,40]. On top of that, introducing a correction factor is an option.

A key finding from Figure 9 is that with both formulations, higher printlet masses were obtained for all volumes examined than theoretically calculated. The calculation of theoretical printing masses was carried out under the assumption of full object infill by utilization of the blend components’ proportional true densities. This indicates slight over-extrusion, which is a typical 3D printing phenomenon where more polymer melt is dispensed than needed to create the object. Dimensional inaccuracies, layer drooping, oozing, or blobbing can accompany over-extrusion even though these effects have not been observed in the current study and appropriate tablets were obtained [41,42,43]. Moderate and particularly constant over-extrusion is a purely technical challenge that can be solved by optimizing printing parameters, such as extrusion speed, nozzle moving speed, or the object’s layer height [41,42,43]. As long as over-extrusion takes place in a constant manner, the introduction of a volume–mass correction factor can be considered once more.

### 3.6. Uniformity of Dosage Units

Investigation of the uniformity of dosage units is crucial in order to guarantee consistency of API content within the batches of 3DP tablets. The results of the mean API contents and acceptance values (AV) are summarized in Table 4. According to Ph. Eur. Monograph 2.9.40 “uniformity of dosage units”, the AV is required to be below 15 [28]. If the AV is greater than 15, 20 additional dosage units need to be tested. In this case the requirements are met if the final AV of 30 dosage units is less than or equal to 15 and no individual dosage unit content deviates from the reference values by more than 25% [28].

Within each batch, no single dose deviates by more than 15% from the respective mean value. All samples met the pharmacopeial specifications regarding the AV and the capability of the multi-step operation to fabricate tablets within acceptable in-batch drug content variations was therefore proven.

### 3.7. In Vitro Dissolution of Solid Oral Dosage Forms

Figure 10 shows the dissolution profiles of the printed MSN tablets. Independently of the polymer used, a complete drug release was achieved within 60 min. The release rates of both formulations are highly similar. The USP considers a single oral dosage IR if an accumulated drug release of more than 80% is achieved within 30 min [44,45]. The recommendation of the Ph. Eur. specifies a drug release equal to or more than 80% in less than 45 min for a conventional IR tablet [46]. The drug release profiles of both formulations meet the specifications of the European Pharmacopoeia. The obtained dissolution results of both formulations do slightly miss the requirements of the USP for immediate release dosage forms, despite the model drug MSN being freely water soluble [44,47]. A major reason accounting for impeded drug release is the compactness and high density of the printed tablets due to full body infill. The polymer melts leave no air inclusions inside the tablet due to their low viscosity. Arafat et al. showed that the dissolution behavior is strongly dependent on the printlet infill. In their studies, decreasing the infill density by introducing cavities into the print file led to increased dissolution rates [48]. By changing the print geometry, the release can also be accelerated. Goyanes’ et al. previously published research has stated that high surface area to volume ratios result in quicker drug release [49]. These aspects could be used to further tailor the release kinetics of the present printlets.

## 4. Conclusions

In this work, polymer systems were developed applicable for a granules-fed 3D extruder printhead. Both printable placebo blends and drug-containing systems were produced using the carrier polymers KVA64 and EPO. The granules obtained were examined for thermal durability in order to prevent decomposition processes during 3DP. The selected 3DP temperature of 140 °C is considered to be safe in terms of thermal degradation for both formulations. In addition, partial drug miscibility of both blends’ components was perceived. Correlation of viscosity profiles with the printability of the formulations shows that the printing temperature is a crucial parameter for successful extrusion and closely related to melt viscosity attributes. Improper printing temperatures lead to physical damaging or inoperability of the printing channel. All formulations display a limited processability window in regards of melt viscosity. The knowledge gained about the required melt viscosity can be incorporated into the development of future pharmaceutical 3DP formulations. With regard to mass uniformity, every print run of both blends fulfilled the European pharmacopeial requirements. While every single printlet fitted the broader second pharmacopeial specification limit, the majority also met the first uniformity criterion. The Assay of actual drug content in the 3DP tablets proves that neither KVA64/MSN/PEG nor EPO/MSN has undergone major API degradation in the course of multiple processing steps from mixture to final product. The Ph. Eur. monograph testing for uniformity of dosage units was met in all cases. Additionally, the drug release properties of both formulations show that immediate drug release is feasible. By further administering improvements such as print geometry optimization, infill adjustments, or addition of disintegrants, the technology holds potential for providing individualized therapy to hypertension patients. In terms of mass and content uniformity, this research work proves that a process eligible for approval by regulatory pharma authorities is possible.

## Figures and Tables

**Figure 1 pharmaceutics-14-02488-f001:**
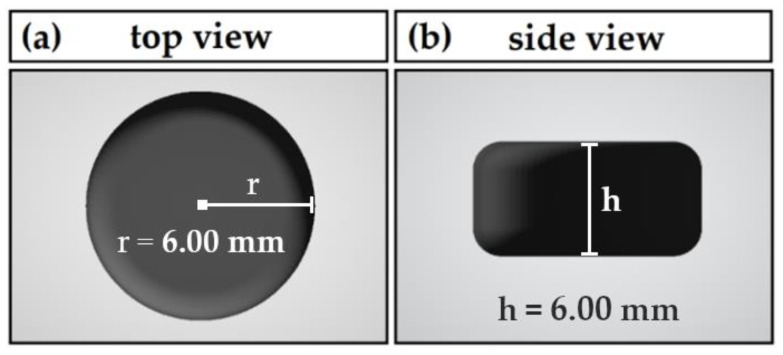
Dimensions and appearance of biplane tablet printlet: (**a**) top view; (**b**) side view.

**Figure 2 pharmaceutics-14-02488-f002:**
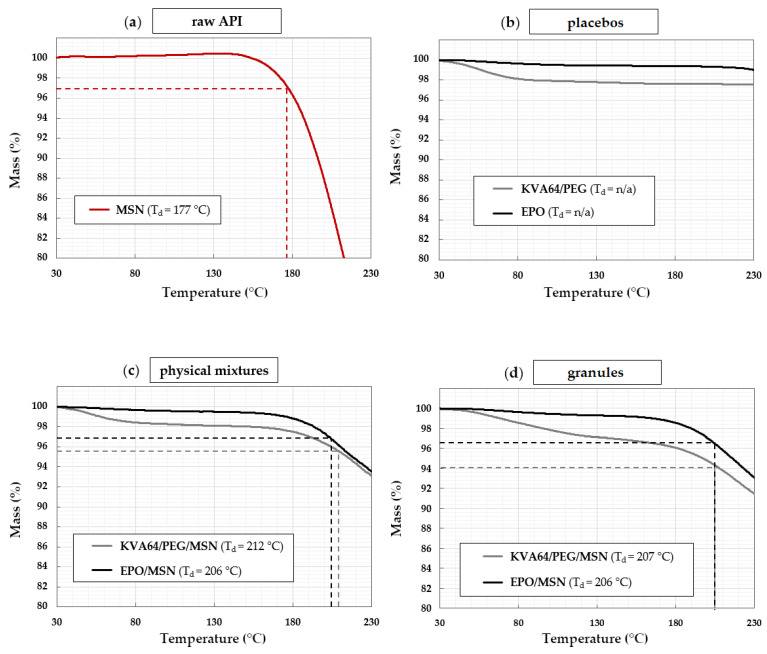
TGA results: (**a**) model drug MSN; (**b**) placebo polymers KVA64/PEG and EPO; (**c**) pre-HME physical mixtures KVA64/PEG/MSN and EPO/MSN; (**d**) post-HME granulated KVA64/PEG/MSN and EPO/MSN.

**Figure 3 pharmaceutics-14-02488-f003:**
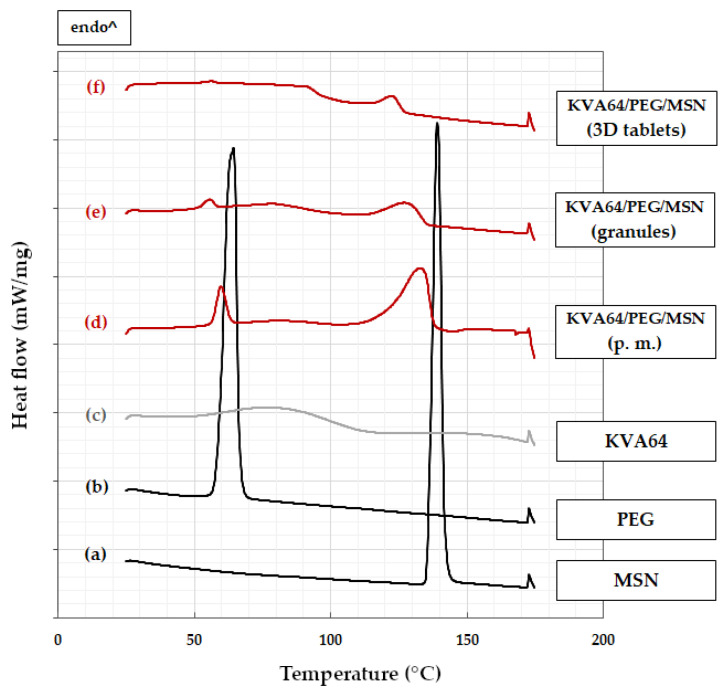
DSC measurements: (**a**) MSN; (**b**) PEG; (**c**) KVA64; (**d**) KVA64/PEG/MSN physical mixture (p. m.); (**e**) KVA64/PEG/MSN granules; (**f**) KVA64/PEG/MSN 3D-printed tablets.

**Figure 4 pharmaceutics-14-02488-f004:**
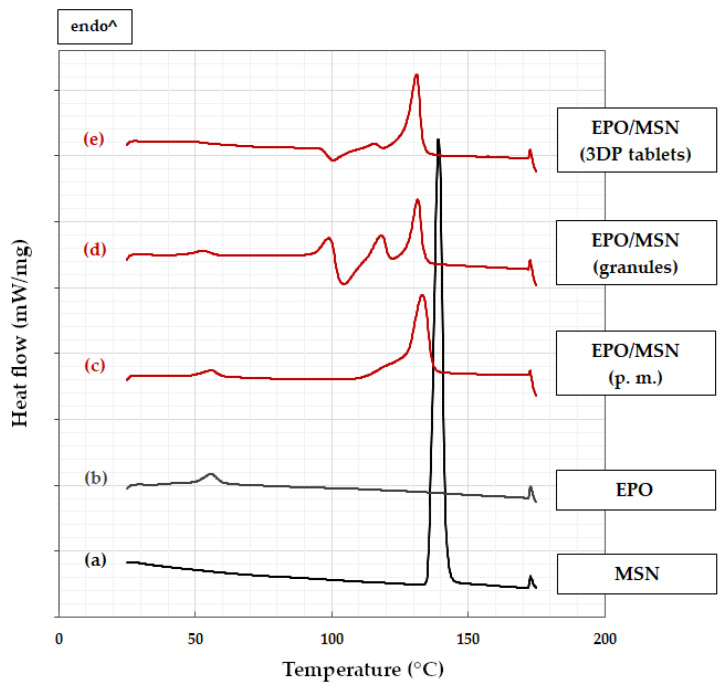
DSC measurements: (**a**) MSN; (**b**) EPO; (**c**) EPO/MSN physical mixture (p. m.); (**d**) EPO/MSN granules; (**e**) EPO/MSN 3D-printed tablets.

**Figure 5 pharmaceutics-14-02488-f005:**
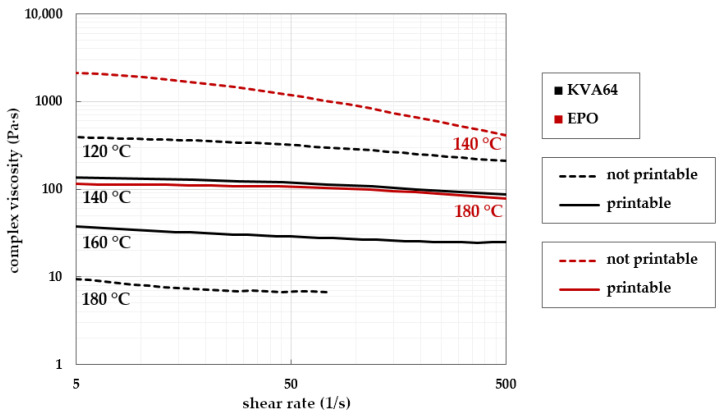
Complex viscosities of placebo formulations KVA64/PEG (black lines) and EPO (red lines) in relation to shear rate monitored at different temperatures; temperatures for successful print runs are symbolized by continuous lines, unsuitable temperature runs in dotted lines.

**Figure 6 pharmaceutics-14-02488-f006:**
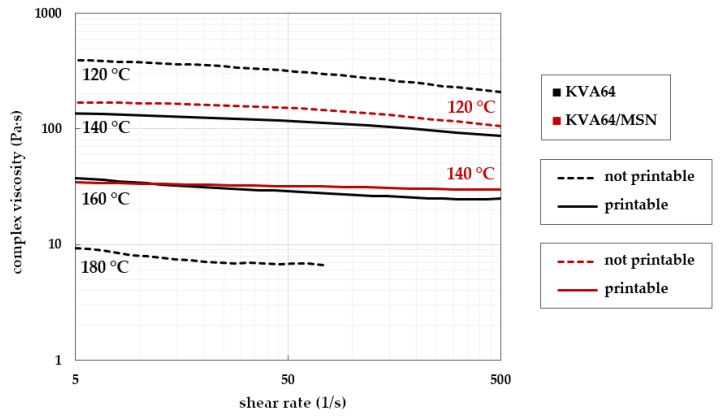
Complex viscosities of placebo formulation KVA64/PEG (black lines) and KVA64/PEG/MSN (red lines) in relation to shear rate monitored at different temperatures; temperatures for successful print runs are symbolized by continuous lines, unsuitable print runs in dotted lines.

**Figure 7 pharmaceutics-14-02488-f007:**
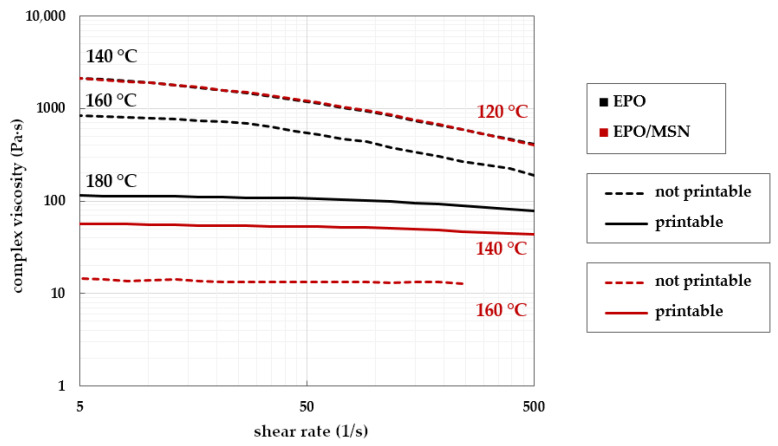
Complex viscosities of placebo formulation EPO (black lines) and EPO/MSN (red lines) in relation to shear rate monitored at different temperatures; temperatures for successful print runs are symbolized by continuous lines, unsuitable print runs in dotted lines.

**Figure 8 pharmaceutics-14-02488-f008:**
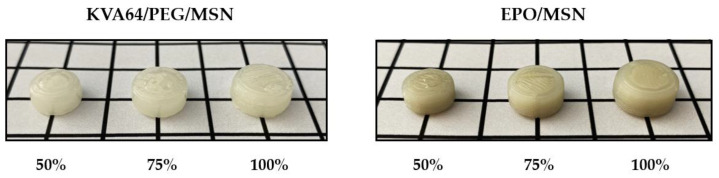
Sample 3DP tablets obtained from formulations KVA64/PEG/MSN and EPO/MSN. The volume scaling factors 100%, 75%, and 50% refer to the standard printlet file mentioned.

**Figure 9 pharmaceutics-14-02488-f009:**
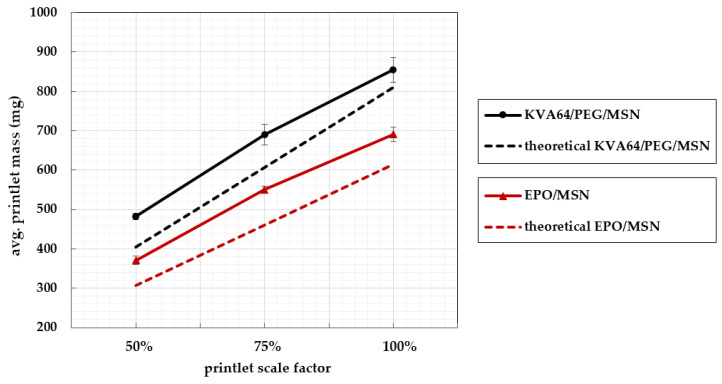
KVA64/PEG/MSN and EPO/MSN’s mean printlet masses with standard deviations versus the printlet volume scale factors 100%, 75%, and 50%.

**Figure 10 pharmaceutics-14-02488-f010:**
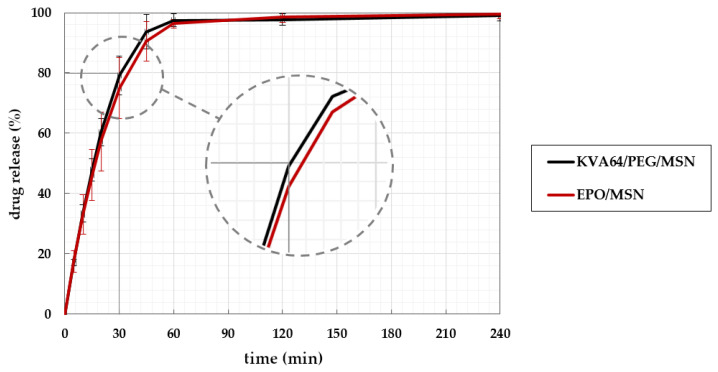
In vitro drug release of the model drug in KVA64/PEG/MSN (black) and EPO/MSN (red) 3DP printlets. The USP criterion for IR is highlighted along the x- and y-axis [44].

**Table 1 pharmaceutics-14-02488-t001:** Compositions and designations of the formulations consisting of Metoprolol Succinate (MSN), Kollidon VA64 (KVA64), Eudragit E PO (EPO), or Lipoxol^®^ 6000 (PEG).

Formulation	MSN (% *w*/*w*)	KVA64 (% *w*/*w*)	EPO (% *w*/*w*)	PEG (% *w*/*w*)
KVA64/PEG	-	70	-	30
KVA64/PEG/MSN	25	65	-	5
EPO	-	-	100	-
EPO/MSN	25	-	75	-

**Table 2 pharmaceutics-14-02488-t002:** Twin-screw tabletop HME parameters for the production of selected polymer blends.

	KVA64/PEG	KVA64/PEG/MSN	EPO	EPO/MSN
Extrusion T. (°C)	100	100	140	140
Torque (N·m)	2.0	2.2	2.4	2.1
Screw speed (rpm)	100	100	100	100
3DP T. (°C)	140	140	180	140

**Table 3 pharmaceutics-14-02488-t003:** Comparison of average printlet mass, standard deviation, and mass uniformity limits for formulations KVA64/PEG/MSN and EPO/MSN.

Formulation	Printlet Scale Factor ^1^	Mean Mass [mg]	±SD	First Ph. Eur. Limit ^2^	Second Ph. Eur. Limit ^3^
KVA64/PEG/MSN	50%	482	1.75%	10/10	10/10
	75%	691	3.73%	9/10	10/10
	100%	854	4.34%	9/10	10/10
EPO/MSN	50%	371	2.92%	9/10	10/10
	75%	550	1.65%	10/10	10/10
	100%	691	2.62%	10/10	10/10

^1^ identical printlet regarding geometry ratio with scale factor relating solely to the objects volume. ^2^ allowing a maximum of 5% deviation from average mass [27]. ^3^ allowing a maximum of 10% deviation from average mass [27].

**Table 4 pharmaceutics-14-02488-t004:** Drug content uniformity of MSN in KVA64/PEG/MSN and EPO/MSN granules and tablets.

	KVA64/PEG/MSN		EPO/MSN	
	Granules	3DP Tablets	Granules	3DP Tablets
Mean API content (%) ^1^	96.9 ± 1.5	96.1 ± 1.7	99.7 ± 1.8	98.3 ± 1.3
Acceptance value	5.2	6.5	4.2	3.3

^1^ corresponding to the expected drug content; (*n* = 10).

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
