# Peer review of "Novel Approach to Pharmaceutical 3D-Printing Omitting the Need for Filament—Investigation of Materials, Process, and Product Characteristics"

_pharmaceutics, 2022, doi:10.3390/pharmaceutics14112488_

Round 1

Reviewer 1 Report

In the present manuscript, the authors deal with a novel printerhead to be used in 3D printing of tablets in order to avoid the need for filament manufacturing, as required by the well-known Fused Deposition Modeling (FDM) technique. Overall, the article is well-written and a lot of data were collected and thoroughly commented. Although the 3D printed dosage forms proposed by the authors is not new form the pharmaceutical perspective (i.e. immediate release tablets), their materialistic approach to formulation characterization would represent quite a novelty in the sector. For this reason, pharmaceutical scientists working in the field would benefit from the availability of this manuscript. However, a few minor improvements would be beneficial to the text before publication, as summarized below.

1) Given the main topic of the present article, I would suggest the authors to expand their Introduction to better cover the FDM technique including the most recent literature references in this respect (e.g. Krueger et al., J. Control. Release, 2022, 351: 444-455; Govender et al., Adv. Drug Deliv. Rev., 2021 177: 113923; Parulski et al., Adv. Drug Deliv. Rev., 2021, 175: 113810). In fact, especially the first sentences are just referred to 3D printing in general and its potential towards personalization. Doing so, this Section results quite old as most of articles already published in the scientific literature report the same considerations. By way of example, besides personalization of the final system, FDM allowed many researchers to evaluate the feasibility of very complex dosage forms/devices, so I would advise the authors to include this aspect to give to their Introduction a diverse spin from what is already available (e.g. Berger et al., J. Control. Release, 2022, 348: 870-880; Zhang et al., Int. J. Pharm., 2022, 625: 122140; Melocchi et al., Phramaceutics, 2021, 13: 759; Menegatou et al., AAPS PharmSciTech, 2022, 23: 205; Zhang et al., Int. J. Pharm., 2022, 624:121972). At the same time, to better understand why the printing mechanism here proposed would be truly advantageous, the criticalities of FDM should better highlighted before citing the DiHeSys employed;

2) At line 66 it was stated: “while filament-fed additive manufacturing is already known and investigated in literature, feeding with granules is a novelty in the pharmaceutical sector”. Actually, other research groups have already proposed the idea to overcome this FDM limitation by using granules as starting materials. I would suggest to revise the sentence accordingly and to cite these works (e.g. Fanous et al., Int. J. Pharm., 2020, 578: 119124; Goyanes et al., Int. J. Pharm., 2019, 567:118471). Also, the authors said that “granules are ready-to-use pharmaceutical polymer formulations that hold particular active ingredients”. However, drug-embedded granules do not represent ready-to-use pharmaceutical polymer formulations as they are generally prepared by hot melt extrusion (HME) starting from powder formulations. In this respect, the authors themselves needed to take advantage of HME to prepare they own granules to be loaded into the equipment available. Even in this case, I strongly suggest a revision of the text;

3) In the aim, the authors should better highlight wihich is the expected release performance for their final printed systems (i.e. immediate release dosage forms) and to justify both the choice of polymers (i.e. promptly soluble and soluble in acidic medium) and of the dissolution conditions employed (i.e. HCl);

4) In Figure 1 dimensions of the tablets are missing.

Author Response

Dear editor,

please find the attached PDF document for a detailed point-to-point response to your comments.

Kind Regards,

The Authors

Reviewer 2 Report

The authors prepared a manuscript in which printlets were produced by 3D printing using two types of supports for metoprolol succinate. 41 references were used for the manuscript. The figures and tables are informative.

A few notes:

- What was the batch size during the production of the powder mixture?

- Fig. In the case of 2., do not use a comma for the values of the Y-axis, but a period.

- In Table 1,  is more correct than specifying the mass percentage; use of %; w/w.

Author Response

(The authors gave the same response as above.)

Reviewer 3 Report

The paper presents interesting example of novel 3D printing technique of tablets using hot melt extrusion. The concept has interesting perspectives from the point of view of scalability - hot melt extrusion belongs to standard manufacturing technologies in the pharmaceutical industry. The Authors presented an interesting example of its application. Although some specific questions should be clarified prior publication. 

1. Is the MSN thermally stable? Why the content of degradation products was not tested during the study?

2. Please explain the term over-extrusion. What does it mean in practice?

3. How the energy distribution was monitored (if at all) during the process. 

4. What was the internal structure of tablets? Is there any density differences between top and bottom of tablet? 

Author Response

(The authors gave the same response as above.)

Reviewer 4 Report

This is very interesting paper which demonstrated successful 3d printing of tablets using printer which does not require filaments and therefore is more suitable for pharmaceutical use. In general, the manuscript is well written, there are only several suggestions to the authors how to further improve it.

1. Since this printer have not been widely used yet in pharmaceutical research, it will be useful to provide its schematic drawing.

2. Line 238-245. The presence of drug partially in amorphous state might induce variations in drug dissolution rate upon storage due to crystallization. This is usually not problem for highly soluble drug, please check if there are some data in the literature regarding solubility difference between amorphous and crystalline form of this API.

3. Exothermic events on the DSC thermograms should be interpreted. This may be recrystallization of amorphous portion of API.

4. Conclusion is too long. It should be shorten just to include important findings from the study and directions for further research.

Author Response

(The authors gave the same response as above.)

Round 2

Reviewer 3 Report

The explanations submitted by Authors are satisfactory.